# Transport in helical Luttinger liquids in the fractional quantum Hall regime

Ying Wang[1], Vadim Ponomarenko[1,2], Zhong Wan [1,6], Kenneth W. West[3], Kirk W. Baldwin[3], Loren N. Pfeiffer[3], Yuli Lyanda-Geller[1,4✉] & Leonid P. Rokhinson [1,4,5✉]

Domain walls in fractional quantum Hall ferromagnets are gapless helical one-dimensional channels formed at the boundaries of topologically distinct quantum Hall (QH) liquids. Naïvely, these helical domain walls (hDWs) constitute two counter-propagating chiral states with opposite spins. Coupled to an s-wave superconductor, helical channels are expected to lead to topological superconductivity with high order non-Abelian excitations[1–3]. Here we investigate transport properties of hDWs in the $\nu = 2/3$ fractional QH regime. Experimentally we found that current carried by hDWs is substantially smaller than the prediction of the naïve model. Luttinger liquid theory of the system reveals redistribution of currents between quasiparticle charge, spin and neutral modes, and predicts the reduction of the hDW current. Inclusion of spin-non-conserving tunneling processes reconciles theory with experiment. The theory confirms emergence of spin modes required for the formation of fractional topological superconductivity.

[1] Department of Physics and Astronomy, Purdue University, West Lafayette, IN, USA. [2] Ioffe Physico-Technical Institute, Saint-Petersburg, Russia. [3] Department of Electrical Engineering, Princeton University, Princeton, NJ, USA. [4] Birck Nanotechnology Center, Purdue University, West Lafayette, IN, USA. [5] Department of Electrical and Computer Engineering, Purdue University, West Lafayette, IN, USA. [6] Present address: Department of Chemistry and Biochemistry, University of California, Los Angeles, CA, USA. ✉email: yuli@purdue.edu; leonid@purdue.edu

Gapless chiral edge states, a hallmark of the quantum Hall effect (QHE), are formed at the boundaries of the two-dimensional (2D) electron liquid. These states are protected due to their topological properties; their spatial separation suppresses backscattering and insures precise quantization of the Hall conductance over macroscopic distances[4]. Symmetry-protected topological systems can support spatially coexisting counter-propagating states. For example, in 2D topological insulators, time reversal symmetry insures orthogonality of Kramers doublets[5–7]; in graphene, the conservation of angular momentum prevents backscattering in the quantum spin Hall effect regime[8]. Local symmetry protection is not as robust as spatial separation in the QHE and, as a result, helical domain walls (hDWs) have finite scattering and localization lengths. Helical states can be also engineered by arranging proximity of two counter-propagating chiral states with opposite polarization, e.g., in electron-hole bilayers[9] or double quantum well structures[10], where local charge redistribution between two quantum wells creates two counter-propagating chiral states at the boundary of quantum Hall liquids with different filling factors. In the latter system, spatial separation into two quantum wells suppresses the interchannel scattering, and transport in each chiral channel is found to be ballistic over macroscopic distances.

An intriguing possibility to form helical channels in the interior of a 2D electron gas is to induce a local quantum Hall ferromagnetic transition. In the integer QHE regime, scattering between overlapping chiral edges from different Landau levels is suppressed due to the orthogonality of the wavefunctions, but spin–orbit interaction mixes states with opposite spins and opens a small gap in the helical spectrum[11–13]. In the fractional quantum Hall effect (FQHE) regime, an electrostatically-controlled transition between unpolarized (u) and polarized (p) $\nu = 2/3$ states results in the formation of a conducting channel at the boundary between u and p regions (filling factor $\nu^{-1} = B/n\phi_0$, where B is an external magnetic field, $\phi_0 = h/e$ is a flux quanta and n is electron density). Superficially, a transition between u and p states in the bulk can be understood as a crossing of two composite fermion energy states with opposite spins polarization[10,14,15]. Within this model, the hDW at the u-p boundary consists of two counter-propagating chiral states with opposite spin and fractionalized charge excitations, and presents an ideal platform to build

fractional topological superconductors with parafermionic and Fibonacci excitations[1,2,16–19]. Highly correlated $\nu = 2/3$ state exhibits rich physics beyond an oversimplified model of $\nu^* = 2$ integer QHE for weakly interacting composite fermions and includes observation of upstream neutral modes[20–23], short-range upstream charge modes[24], and a crossover from $e^* = 1/3$ to $e^* = 2/3$ charge excitations in shot noise measurements[25]. In the bulk, the spin transition at $\nu = 2/3$ is accompanied by nuclear polarization[26–28] indicating spin-flip processes in the 2D gas, a phenomena not seen in bilayer systems[10,29]. Thus, we expect a hDW formed at a boundary between u and p $\nu = 2/3$ states to be more complex than a simple overlap of two noninteracting chiral modes.

Here we study the electron transport in samples where hDWs of different length L are formed by electrostatic gating. Experimentally, in the limit $L \rightarrow 0$ only 11% of the edge current is diverted into the hDW, a number drastically different from the 50% prediction for two noninteracting counter-propagating chiral channels. To address this discrepancy theoretically, we consider tunneling between Luttinger liquid modes[30] through a hDW in the strong coupling limit[31–33] and confirm that results remain the same if hDW is modeled as an extended object. We found that in the presence of a strong inter-edge tunneling edge channels in u and p regions populate unequally, both at the boundary of the 2D gas and within the hDW, forming a number of down- and upstream charge, spin, and neutral modes. For spin-conserving tunneling 1/4 of the incoming charge current is diverted into the hDW, while allowing spin-flip processes further reduces hDW current. Indeed, at high bias currents, we observe an increase in the current carried by the hDW. This indicates the formation of a bottleneck for spin flips due to the Overhauser pumping of nuclei and a crossover from spin-non-conserving to spin-conserving transport.

## Experimental Results

Several devices in a Hall bar geometry with multiple gates have been fabricated in order to study transport through hDWs, Fig. 1 (for heterostructure and fabrication details see Methods). Devices are separated into two regions G1 and G2; electron density n in these regions can be controlled independently by electrostatic

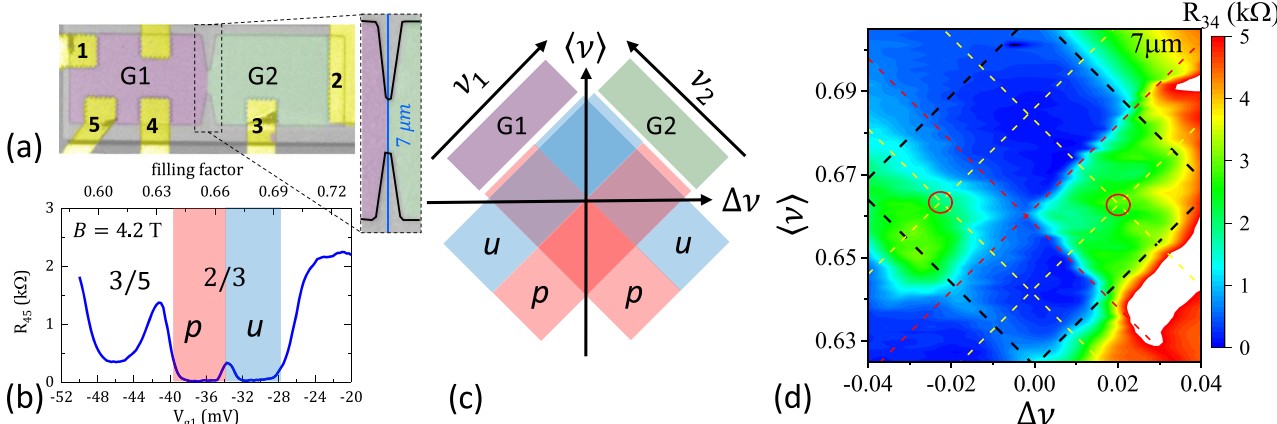

**Fig. 1 Formation of helical domain walls at $\nu = 2/3$. a** A false-color image of a typical device. Yellow regions are ohmic contacts, 2D gas in green and magenta regions is controlled by gates G1 and G2, in the gray area, 2D gas is removed. In the enlarged section, thick black lines outline the mesa boundary and a vertical blue line marks a physical boundary between G1 and G2. **b** Magnetoresistance $R_{45} = (V_5 - V_4)/I$ of a 2D gas is plotted as a function of gate voltage (controlling filling factor $\nu$) at a fixed $B = 4.2$ T. Small peak at $-34$ mV is the phase transition between unpolarized (u) and polarized (p) $\nu = 2/3$ FQHE liquids. **c** A diagram of u and p states as a function of $\nu_1$ and $\nu_2$ under gates G1 and G2; $\langle\nu\rangle = (\nu_1 + \nu_2)/2$ and $\Delta\nu = (\nu_1 - \nu_2)$. **d** Resistance $R_{34} = (V_4 - V_3)/I$ across a 7 $\mu$m-long gates boundary is plotted as a function of $\langle\nu\rangle$ and $\Delta\nu$. The black square outlines the $\nu = 2/3$ region, red lines mark u-p transitions and yellow lines mark centers of u and p regions.

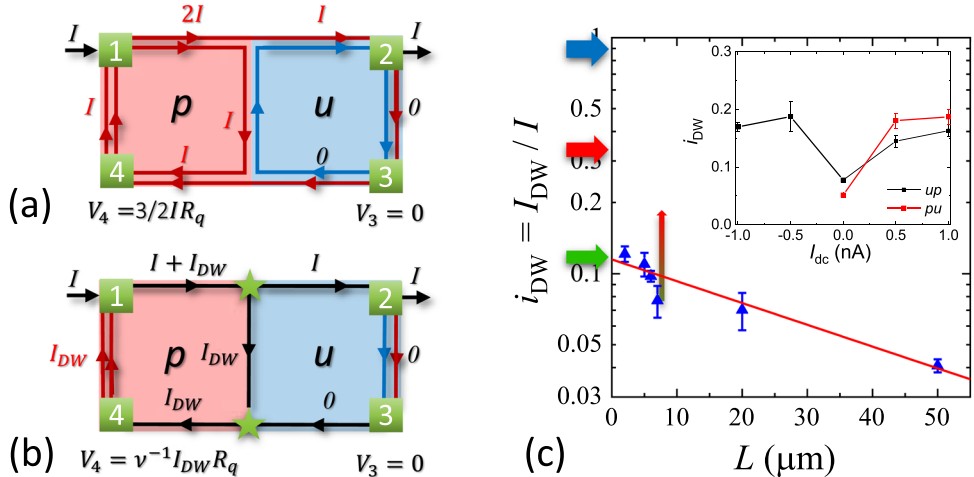

**Fig. 2 Conduction of helical domain walls. a** A simplified picture of noninteracting chiral edge modes at $\nu = 2/3$. An inner spin-up edge (red) in $p$ state carries current $I$ while a spin-down edge (blue) in $u$ state carries no current. **b** Charge conservation and chirality of edge states set the potential $V_4 = \nu^{-1}I_{DW}R_q$ to be proportional to the current $I_{DW}$ diverted via the helical domain wall. **c** Scaling of the domain wall current $i_{DW} = I_{DW}/I$ with hDWs length $L$. The values are averaged between $up$ and $pu$ states (within red circles in Fig. 1d), error bars are standard deviations. Red line is a fit to an exponential decay with the $L = 0$ value $i_{DW}^0 = 0.115$ and the decay length $L_0 = 47\,\mu m$. Arrows indicate $i_{DW}^0$ values expected for naive noninteracting edge model (blue) and Luttinger liquid model in the absence of spin-flip (red) and at spin-flip probability $r = 3/4$ (green), see text for details. Vertical arrow marks $i_{DW}$ shift when $I_{dc} = 1\,nA$ is applied. In the inset $i_{DW}$ dependence on large external dc current is plotted for $7\,\mu m$ hDW for $up$ and $pu$ gates configuration right after the dc current is applied and before a measurable build-up of nuclear polarization.

gates. In the IQHE regime when filling factors $\nu$ under gates $G1$ and $G2$ differ by one, a single chiral channel is formed along the gates boundary and resistance $R_{34}$ measured across the boundary is either quantized or zero depending on the sign of the filling factor gradient and direction of $B$ (see Supplementary Note 2). Likewise, a chiral channel is formed when the gates boundary separates two different FQHE states.

Fractional QHE states can be understood as integer QHE states for composite fermions in a reduced field, $\nu = \nu^*/(2\nu^* \pm 1)$, where $\nu^*$ is the filling factor for composite fermions[34]. The energy separation between composite fermions levels depends on the competition between charging energy $E_c \propto \sqrt{B}$ and Zeeman energy $E_Z \propto B$, and the two lowest energy levels with opposite spins 0-down and 1-up cross at finite $B^* > 0$ due to different field dependencies. When the level crossing occurs within the $\nu = 2/3$ plateau ($\nu^* = 2$ for composite fermions), the energy gap for quasiparticle excitations vanishes providing a mechanism for charge backscattering and, hence, at $B = B^*$ resistance of the 2D gas is no longer zero. In our devices, it is possible to control $B^*$ by electrostatic gating, and a small peak within the 2/3 plateau in Fig. 1b is a quantum Hall ferromagnetic transition between polarized ($p$) and unpolarized ($u$) regions.

Independent control of filling factors $\nu_1$ and $\nu_2$ under G1 and G2 divides the 2/3 region into four quadrants $uu$, $pp$, $up$, and $pu$, where the first letter corresponds to a polarization of the state under G1 and the second corresponds to polarization under G2. Within the Landauer–Büttiker formalism[35,36] resistance $R_{34} = (1/\nu_1 - 1/\nu_2)R_q$ should be zero for all combinations of polarizations under the gates since quantum numbers $\nu_1 = \nu_2 = 2/3$ for both $u$ and $p$ states (here $R_q = h/e^2$, $h$ is the Plank's constant and $e$ is an electron charge). Experimentally $R_{34}$ is found to be vanishingly small in $uu$ and $pp$ quadrants, consistent with a single topological state being extended over the whole device. $R_{34} > 0$ in $up$ and $pu$ quadrants indicates backscattering between edge channels and, combined with zero longitudinal resistance under G1 and G2, reflects the formation of a conducting channel along the gates boundary. Unlike resistance measured across chiral channels formed, e.g., between $\nu = 2/3$ and $\nu = 3/5$ FQHE states (see Supplementary Note 2), the resistance measured across the

boundary of $u$ and $p$ quantum liquids at $\nu = 2/3$ shows almost no dependence on the direction of the external magnetic field and density gradient, consistent with the formation of a hDW[37].

Protection of helical states from backscattering and localization is weaker for spatially separated chiral edge states, and conduction of hDWs is length-dependent. In Fig. 2a fraction of the external current $I$ that flows through the hDW, $i_{DW} = I_{DW}/I$, is plotted as a function of hDW length $L$. $i_{DW}$ is found to decrease exponentially with $L$, $i_{DW} = i_{DW}^0 \exp[-L/L_0]$, with a characteristic length $L_0 = 47\,\mu m$. The value of $i_{DW}^0$ corresponds to the transport through a ballistic hDW in the absence of localization. Within a simplified model of $\nu = 2/3$ edge states consisting of equally populated $1/3 + 1/3$ chiral modes and no interaction between chiral channels with opposite spin polarization (Fig. 2a), one expects $i_{DW}^0 = 1$ (marked by a blue arrow in Fig. 2c), an order of magnitude larger than the experimentally observed value (marked by a green arrow). Note that transport through helical modes formed in double quantum well structures are well described by this simple model of weakly interacting chiral states[10].

## Theory

An isolated hDW at a boundary of $p$ and $u$ phases was studied in refs. [19,37], where disk and torus geometries were employed to avoid physical edges of the sample and coupling of domain wall modes to these edges. Analytical model and numerical results indicate the existence of modes with opposite velocities and spins within the hDW region, a prerequisite for generating topological superconductivity. No neutral or spin modes appear within the K-matrix Luttinger liquid approach[30] in these isolated hDW models.

To calculate the scattering of edge modes at a sample boundary by a hDW we need to move beyond an isolated hDW model. A conventional starting point for chiral edge states description are two filled $\Lambda$-levels of composite fermions with equal or opposite spin in the $p$ and $u$ phases, with edge modes described by densities $\Phi_{p1\uparrow}$, $\Phi_{p2\uparrow}$ and $\Phi_{u1\uparrow}$, $\Phi_{u2\downarrow}$ correspondingly. As shown in Supplementary Note 3, applying unitary transformations, we arrive to the description in terms of the separated charged and neutral modes

$\varphi_{pc} = (\Phi_{p1\uparrow} + \Phi_{p2\uparrow})/3\sqrt{2}$ and $\varphi_{pn} = (\Phi_{p1\uparrow} - \Phi_{p2\uparrow})/\sqrt{2}$ for the $p$ phase, and separated charged and spin modes $\varphi_{uc} = (\Phi_{u1\uparrow} + \Phi_{u2\downarrow})/3\sqrt{2}$ and $\varphi_{us} = (\Phi_{u1\uparrow} - \Phi_{u2\downarrow})/\sqrt{2}$ for the $u$ phase, equivalent to the composite fermion description. The Luttinger liquid action for $\nu = 2/3$ edge states, e.g., for the $u$ phase in terms of separated charge and spin modes, reads

$$S = \frac{1}{4\pi} \int dt \int dx \left[ -3\partial_x \varphi_{uc}(\partial_t + v_c\partial_x)\varphi_{uc} + \partial_x \varphi_{us}(\partial_t - v_s\partial_x)\varphi_{us} \right], \quad (1)$$

where $v_c$ and $v_s$ are velocities of charge and spin modes, correspondingly. The spin mode determines the spin current in $u$ phase. The $p$ phase is described by a similar action, in which the neutral $\varphi_{pn}$ mode that determines a difference in the occupation of the first two composite fermion $\Lambda$-levels enters instead of $\varphi_{us}$, the velocity of neutral mode $v_n$ enters instead of $v_s$, and $\varphi_{pc}$ appears instead of $\varphi_{uc}$. These actions coincide with the Kane, Fisher, and Polchinsky[38] Luttinger liquid action expressed in terms of the charge and neutral fields, see also refs. [30,39], in the absence of tunneling between composite fermion modes due to impurity scattering. Indeed, at the edges of the $u$ phase such tunneling is forbidden in the absence of spin-flip processes, and $u$ phase exhibits spin-charge separation with pure spin and charge modes moving in opposite directions[40] as in Eq. (1). To treat both phases, which exhibit quantization of Hall resistance $3/2h/e^2$ and similar longitudinal resistance characteristics, on equal footing, we assume that no scattering between different modes occurs along edges in the $p$ phase also. At the same time, we show in Supplementary Note 3 that scattering between quasiparticle modes inside the domain wall does not alter measurable currents. In both phases quantization of the Hall resistance and charge and neutral (spin) mode separation is a consequence of the long-range Coulomb interaction[30].

We generalize the known solution for tunneling[41] of the fractional QHE modes through the point contact to consideration of tunneling through finite length domain wall between $p$ and $u$ phases. Point contact tunneling carried by electrons with the same spin can be described by the tunnel Hamiltonian

$$\mathcal{H}_T = -\tilde{t}\cos\left(\Phi_{p1\uparrow} - \Phi_{u1\uparrow}\right) = -\tilde{t}\cos\left(\frac{1}{\sqrt{2}}\left[3\varphi_{pc} - \varphi_{pn} - 3\varphi_{uc} + \varphi_{us}\right]\right). \quad (2)$$

Mapping of an edge-hDW-edge structure onto one-dimensional bosonic modes $\varphi(x)$ is shown schematically in Fig. 3a. In general, hDW and each region outside of the hDW may contain up to eight

bosonic modes $\varphi_{\alpha\beta}^{\rightleftarrows}(x)$, where $\alpha = \{p, u\}$, $\beta = \{c, n/s\}$ and the superscript $\{\leftarrow, \rightarrow\}$ indicates the projection of the velocity $v_\beta$ on the x-axis. However, the chirality of edge channels reduces the number of bosonic modes to four. It is convenient to consider two outgoing charge modes $\varphi_{pc}^\rightarrow(x_2)$ and $\varphi_{uc}^\leftarrow(x_1)$ and two outgoing spin/neutral modes $\varphi_{pn}^\leftarrow(x_1)$ and $\varphi_{us}^\rightarrow(x_2)$.

In the strong coupling limit $\tilde{t} \to \infty$, charge, neutral, and spin currents can be found by imposing the following boundary conditions on bosonic fields right outside of the hDW $[x_1, x_2]$:

$$\begin{pmatrix} \varphi_{pc}^\rightarrow(x_2 + 0) \\ \varphi_{us}^\rightarrow(x_2 + 0) \\ \varphi_{uc}^\leftarrow(x_1 - 0) \\ \varphi_{pn}^\leftarrow(x_1 - 0) \end{pmatrix} = \frac{1}{4}\begin{pmatrix} 1 & -1 & 3 & 1 \\ -3 & 3 & 3 & 1 \\ 3 & 1 & 1 & -1 \\ 3 & 1 & -3 & 3 \end{pmatrix}\begin{pmatrix} \varphi_{pc}^\rightarrow(x_1 - 0) \\ \varphi_{us}^\rightarrow(x_1 - 0) \\ \varphi_{uc}^\leftarrow(x_2 + 0) \\ \varphi_{pn}^\leftarrow(x_2 + 0) \end{pmatrix}, \quad (3)$$

which connect all incoming modes at the right side and outgoing modes at the left side of the equation. Our principal result is that imposing strong coupling boundary conditions in a general case of a domain wall of finite length results in the same currents flowing outside the domain wall as for the models of single-junction connecting edges on opposite sample boundaries, two junctions on opposite edges, and two junctions with scattering between same spin modes in between. Inside the domain wall, the chiral evolution of modes is controlled by the average voltage shifts at their corresponding boundaries. In the presence of voltage $V$, the only incoming mode changing due to charge injection in the $p$ phase is $\varphi_{pc}(x_1)$, characterized by an average induced current $\bar{j} = \frac{e^2 V}{3\pi h}$.

When spin-flip processes are absent, it is convenient to discuss the results in terms of currents carried by $\Phi_{p1}$ ($\Phi_{u1}$) and quasiparticle $\chi_{p2} = (\varphi_{pc} - \varphi_{pn})/\sqrt{2}$ ($\chi_{u2} = (\varphi_{pc} - \varphi_{ps})/\sqrt{2}$) modes. We show that $\Phi_{p1\uparrow}$ ($\Phi_{u1\uparrow}$) modes propagate along the edges of the 2D gas and do not enter the domain wall, while $\chi_{p2}$ and $\chi_{u2}$ modes flow along the boundaries of $p$ and $u$ phases correspondingly, including inside the domain wall, as shown schematically in Fig. 3b. Notable features of our solution are unequal distribution of carried currents between the modes caused by strong coupling to the domain wall and the presence of spin current along the edge of the $u$ phase for $x > x_2$. The total current flowing along the hDW $I_{DW} = 1/4(I + I_{DW})$ or $i_{DW} = 1/3$. This value is three times

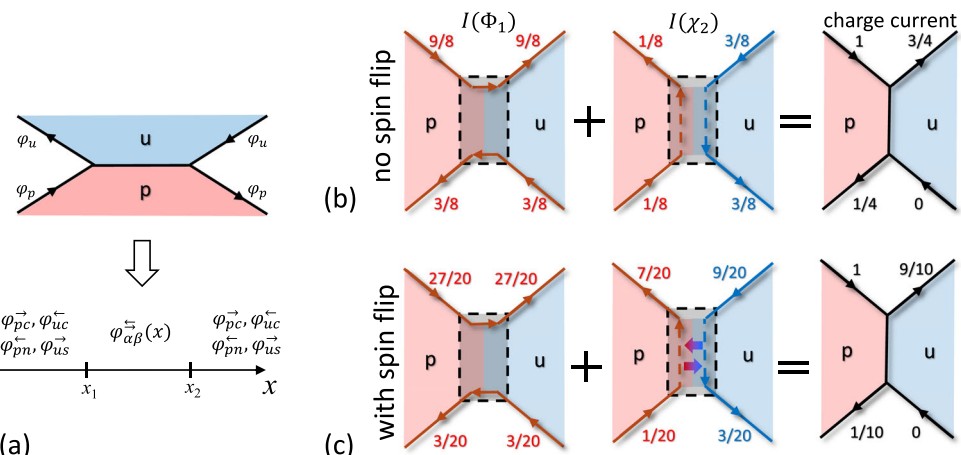

**Fig. 3 Schematic representation of currents. a** Mapping of bosonic modes $\varphi_\alpha$ along sample edges onto a 1D Luttinger model modes $\varphi_{\alpha\beta}^{\rightleftarrows}(x)$ for a domain wall with length $L = x_2 - x_1$. Subscripts $\alpha = \{p, u\}$ label polarized and unpolarized phases, $\beta = \{c, n, s\}$ is for charge, neutral, and spin modes and an arrow in the superscript specifies projection of the mode's group velocity. Arrows on the edges define the chirality of edge channels. **b, c** Visualization of currents due to the propagation of $\Phi_1$ and $\chi_2$ modes without spin flips (**b**) and in the presence of spin flips with the probability $r = 3/4$ (**c**). Red (blue) mode color indicates a spin-up (spin-down) polarization. Numbers indicate the fraction of the incoming current carried by the mode. Arrows correspond to directions of currents carried by corresponding modes.

larger than the experimentally measured $i_{DW}$ and is indicated by a red arrow in Fig. 2c.

In 2D gases formed in GaAs heterostructures spin transition at $v = 2/3$ is accompanied by a dynamic nuclear spin polarization[42]. Its main mechanism is the hyperfine coupling of electron and nuclear spins, which for QHE plateaus is usually suppressed due to a large difference between electron and nuclear Zeeman splitting. Near the $u$-$p$ phase transition, however, electronic states with spin-up and spin-down are almost degenerate, enabling the hyperfine coupling. This spin-flip mechanism can lead to the scattering between $\chi_{p2}$ and $\chi_{u2}$ modes propagating inside the domain wall, see Supplementary Note 3. Notably, $\Phi_1$ modes still propagate along the 2D gas boundary and do not enter the domain wall. However, conservation of total current carried by $\Phi_1$ and $\chi_2$ modes results in the current redistribution between the modes. The ratio characterizing the domain wall current $i_{DW}(r)$ becomes a function of the spin-flip probability $r$ and changes continuously between 1/3 for $r = 0$ and zero for $r = 1$. Experimentally measured values correspond to $r \approx 3/4$, corresponding currents are shown schematically in Fig. 3c.

To test the role of spin flips it is possible to pass a large dc current and polarize nuclei in the vicinity of the tri-junction. Saturation of nuclear spin polarization is expected to create a bottleneck for electron spin flips and disable charge transfer between two $\chi_2$ bosonic modes with opposite polarization. Indeed, application of $I_{dc} > 0.5$ nA results in approximately a threefold increase of $i_{DW}$, as shown in the inset in Fig. 2. A corresponding shift of $i_{DW}$ for the 7 $\mu$m hDW is shown with a vertical arrow on the main plot. This shift is consistent with the triple current increase expected for the crossover from a spin-flip-dominated to a no-spin-flip transport.

## Methods

Devices are fabricated from GaAs/AlGaAs inverted single heterojunction heterostructures with electron gas density $0.9 \cdot 10^{11}$ cm$^{-2}$ and mobility $5 \cdot 10^6$ cm$^2$/Vs. Details of heterostructure design can be found in ref. [43], these heterostructures demonstrate efficient electrostatic control of the spin transition at $v = 2/3$, see ref. [37]. Devices are patterned in a Hall bar geometry using e-beam lithography and wet etching, the photograph of a typical sample is shown in Fig. 1. Devices are divided into two regions by semitransparent gates (10 nm of Ti), the gates are separated from the surface of the wafer and from each other by 50 nm of Al$_2$O$_3$ grown by the atomic layer deposition (ALD). Gates boundary is aligned with a 2–50-μm-wide mesa constriction. Ohmic contacts are formed by annealing Ni/Ge/Au 30/50/100 nm at 450 °C for 450 s in $H_2/N_2$ atmosphere. Measurements are performed in a dilution refrigerator at a base temperature of 20 mK using conventional low-frequency lock-in technique with excitation current $I_{ac} = 100$ pA. A 2D electron gas is formed by shining a red LED at 4 K. More details on the 2D gas preparation and sample characterization can be found in Supplementary Note 1.

## Data availability

The data that support the findings of this study are available from the corresponding author upon reasonable request.

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

## Acknowledgements

The experimental part of the work is supported by NSF award DMR-1836758 (Y.W. and L.P.R.). Theoretical work is supported by the US Department of Energy, Office of Basic Energy Sciences, Division of Materials Sciences and Engineering under Award DE-SC0010544 (Y.L.-G.). Heterostructures development and growth is funded in part by the Gordon and Betty Moore Foundation's EPiQS Initiative, Grant GBMF9615 to L.N.P., and by the NSF MRSEC grant DMR-1420541.

## Author contributions

L.P.R. conceived, Y.W. and Z.W. performed experiments, V.P. and Y.L.-G. developed theory, K.W.W., K.W.B., and L.N.P. developed and grew heterostructure materials. Y.W., V.P., Y.L.-G., and L.P.R. have written the manuscript.

## Competing interests

The authors declare no competing interests.
