## [Peer Review File · Nature Communications]

REVIEWER COMMENTS

Reviewer #1 (Remarks to the Author):

This paper describes the transport characteristics of helical one-dimensional channels composed of two counter-propagating channels with opposite spins. The authors fabricated such a device by attaching spin-polarized and unpolarized domains of fractional QH liquids at $\nu = 2/3$, as previously reported in Ref. 37. A four-terminal measurement was presented where backscattering is allowed through the helical channels. The helical channels were formed by independent control of phase transitions in the neighboring regions and confirmed by observing finite four-terminal resistance. The resistance directly reflects the ratio of the backscattering current to the incident current, from which the helical channels were evaluated. Firstly, the resistance decreases with increasing the channel length, which can be understood with tunneling between the helical channels. Secondly, the resistance increases with increasing DC bias current, which can be explained with nuclear spin polarization. These experimental features were discussed with Luttinger-liquid theory with and without spin flip. The experimental scheme and the results are interesting and valuable for understanding the topologically protected helical channels in a realistic device. However, I would like to comment on a few points before recommending this paper for publication.

In the theory part, the authors derived the scattering matrix Eq. (3) for the strong coupling limit without spin flip to deduce $i_{\text{DW}} = 1/3$ (the red arrow in Fig. 2(c)). This derivation is shown in Supplementary Materials. However, I cannot find a clear reason for $i_{\text{DW}} = 1/7$ with spin flip (the green arrow). A clear explanation is required, as the authors used this to validate the result. Some spin flip process might be considered for the zero-length junctions (i_{DW} at $L = 0$). While hyperfine coupling is present in GaAs, I am wondering it is so significant to reduce i_{DW} from $1/3$ to $1/7$ in the zero-length limit.

The backscattering current decreases with increasing the helical channel length at a relatively long characteristic length of $L_0 = 47 \text{ } \mu\text{m}$. Whereas this mechanism is not clearly stated in the text, this may involve a spin-flip process in the helical channels. If so, this characteristic length may change with nuclear spin polarization. The authors could have distinguished the effects on L_0 and i_{DW}^0 by repeating similar high-bias measurements for several lengths. Otherwise, the two possibilities should be described in the text.

The authors applied a large current to polarize nuclear spins. However, this is not demonstrated because the large current may enhance backscattering in the bulk region. The authors should provide evidence of nuclear spin polarization, for example, long-term hysteresis associated with slow nuclear relaxation or NMR response.

Reviewer #2 (Remarks to the Author):

The present work studies the evolution of the current carried at the boundary of two fractional quantum Hall phases presenting different spin polarization. The measured current deviates from the predicted value calculated in an ideal scattering free case. Developing a novel theory including spin flip processes the authors found an agreement with their measurement.

The paper is well written and follows a clear experimental logic as well as a subsequent theoretical explanation. Nevertheless, the claims being pretty high I think the authors should provide additional data to ensure that the measured current is indeed coming from such a helical domain wall and not from a more trivial mechanism.

For that I would ask for the measurement of the evolution of the longitudinal resistance $R_{\{12,34\}}$ when both gates (G1 and G2) are connected to the same voltage source. This would indicate how transparent is the constriction.

Can the authors provide the same colorplot ($R, \nu, \Delta \nu$) for opposite magnetic fields on the same sample? Contacts (especially for small ones) can have a huge impact on the measured data. Having opposite chirality can eliminate partly such doubts

Also is the tendency with length the same if the author takes the highest signal they get (in the green region) on the color plot (so not around the red circles). Density variations in the Hall bar can shift the real filling factor from the one calculated.

Did the authors measure the evolution of the measured current as a function of the magnetic field. I believe that by tuning the gate they probably can measure it at a much higher magnetic field which would make such states also more robust.

Does the author measure the Fig2c Inset at higher DC current than the one plotted on the graph? A clear saturation of the i_{DW} would give more credit to the theoretical model developed.

Does the authors did any temperature dependence measurement of the current measured

Some additional remarks:

In Fig c the vertical arrow is not explained in the caption so we understand the meaning only at the end of the paper

Luttinger is sometimes named Luttenger

On Fig1 a) a clearer scale bar would help to understand the dimensions. It took me time time to understand that "The bar is 7 micron" does not refer to the Hall bar but to the scale bar

On Fig1 d in the colorbar scale the integer "3" took the liberty to appear twice.

Reviewer #3 (Remarks to the Author):

The paper by Wang et al., "Transport in helical Luttinger liquids in the fractional quantum

Hall regime" presents an interesting study of a promising system which (presumably) realizes two distinct fractional quantum Hall liquids at the same filling fraction, $2/3$, separated by a domain wall. While I applaud the authors for undertaking such a study and find the results quite intriguing, I cannot recommend this paper for publication in its current form. Unfortunately, the presentation is wanting, to the point that I can understand neither the experimental findings nor the theory that supposedly explains those.

Let me begin with the experimental part. First and foremost, how do we know that the two halves of the sample are in the desired FQH states to begin with? I presume there is some evidence pointing

to that but I cannot seem to find it in the paper. I am not even sure we are presented with the evidence that both sides of the device are at $\nu=2/3$. E.g., Figure 1 presents some data on R_{45} and R_{34} , but all three leads appear to be attached to the same bottom edge (on both sides of the constriction). The relation of the measured resistances to the standard two- or four-contact measurements is unclear to me. Specifically, the vanishing R_{34} indicates that both halves of the device are sharing the same charge mode but beyond that, how do we know what state(s) this mode belongs to? Figure S2 of the Supplementary Materials *might* contain some evidence pointing in the right direction but even if so, it is indirect and is not articulated in any easily accessible form. What I would ideally like to see is some “conventional” transport data indicating both the nature and the quality of the underlying FQH states. Specifically, would it be possible to open the constriction and gate the device uniformly first, demonstrating a bulk $\nu=2/3$ state across the entire device through the conventional transport? Then, ideally, I would like to see a transition into a different $\nu=2/3$ state controlled by the electrostatic gating. One should ideally be able to see transport signatures of a transition between the two states, as the bulk gap would vanish. While my request may be too ambitious (it is not obvious to me why it should be, but I am willing to give experimentalists the benefit of the doubt), *some* characterization along these lines should be presented.

Beginning with establishing a uniform state across the entire device and characterizing it also crucial for understanding the role the constriction plays in this experiment. A study of tunnelling across the constriction in the absence of any domain walls would undoubtedly help separate transport signatures that are unique to the formation of a domain wall across the constriction.

The issue of tunnelling across the constriction brings me to another gripe: how is the alleged length of the domain wall L determined experimentally? Since it is formed by electrostatic gating, it would be good to understand how the corresponding gate voltage is converted into microns that are used in Fig. 2(c). (On a related note, it would be helpful to see L marked on one of the figures, perhaps Fig. 1(a).)

The lack of characterization of the device also makes me wonder how the authors can rule out an emergence of some compressible strip between the two quantum Hall states (instead of a microscopically sharp helical domain wall)?

Turning to the theoretical description of the underlying physics, I must say that I also find it wanting. I do not find the presentation particularly helpful and in places it becomes outright confusing.

To begin with, the authors jump between composite fermion (CF) and hierarchical pictures making it hard to follow. Specifically, the spin-polarized and unpolarized $\nu=2/3$ states are described in terms of CF's in both the Introduction and Experimental Results section after which the paper jumps to the hierarchical description in the Theory section. Worse yet, the very first equation describing the action for the hierarchical edge of the polarized state is either wrong or at best mischaracterized in the surrounding texts. Setting aside the wrong overall sign, the action in Eq. (1) is clearly written for

the K-matrix of the form $\text{diag}(1,-3)$, which describes two counterpropagating *charge* modes in the standard hierarchy construction, not “separated charged and neutral modes” as stated in the paper. Compare this equation to Eqs. (1) as well as (7)-(8) of the seminal paper by Kane, Fisher & Polchinski, (Phys. Rev. Lett. 72, 4129 (1994)) outlining the phenomenon of spin-charge separation on such an edge. Curiously, the aforementioned paper is not even cited here, and nor is any other paper by Kane & Fisher – see e.g. their chapter “Edge state transport” in “Perspectives in Quantum Hall Effects” (1996), doi:10.1002/9783527617258.ch4. Just to be clear, I am not trying to make the authors add any citations here; I merely find the lack of these very relevant citations combined with the wrong/confusing statements rather worrisome. And it is not as if the authors felt pressured by the size constraints: in the same section they digress and discuss the inclusion of superconductivity, which is completely irrelevant to the subject at hand.

My point is that the “Theory” section in its present form has hardly helped me understand the physics being studied here. If anything, I found it both sloppy and confusing. (This unfortunate impression is further reinforced by relatively minor, yet adding-up things like a missing spatial derivative in the second term in Eq. (1), mismatching parentheses in Eq. (S2) or missing parentheses in Eq. (S6).)

The sloppiness of the presentation is also evident in writing: “level crossing” requires an article in English; the QH discoverer’s name is *von* Klitzing, not Klitzing, and in any case, purely grammatically, it is either “the von Klizing constant” or “von Klitzing’s constant” (no “the”). Besides, it is simply beyond me why one would use R_q (and call it “the von Klizing constant”) instead of simply using h/e^2 , requiring no explanation or commentary of any kind.

To summarize, I really applaud the authors’ attempt to study this very interesting system and would really like to see results of such a study. Unfortunately, this paper falls short and cannot recommend its publication in its current form.

Transport in helical Luttinger liquids in the fractional quantum Hall regime

Ying Wang, Vadim Ponomarenko, Zhong Wan, Kenneth W. West, Kirk Baldwin, Loren N. Pfeiffer, Yuli Lyanda-Geller, Leonid P. Rokhinson

Response to Reviewers' Comments

Reply to Reviewer #1:

We are happy that the Reviewer finds “The experimental scheme and the results are interesting and valuable for understanding the topologically protected helical channels in a realistic device” and we answer below specific questions raised by the Reviewer.

In the theory part, the authors derived the scattering matrix Eq. (3) for the strong coupling limit without spin flip to deduce $i_{DW} = 1/3$ (the red arrow in Fig. 2(c)). This derivation is shown in Supplementary Materials. However, I cannot find a clear reason for $i_{DW} = 1/7$ with spin flip (the green arrow). A clear explanation is required, as the authors used this to validate the result. Some spin flip process might be considered for the zero-length junctions (i_{DW} at $L = 0$). While hyperfine coupling is present in GaAs, I am wondering it is so significant to reduce i_{DW} from $1/3$ to $1/7$ in the zero-length limit.

We thank the referee for this important question. While it may have been confusing in the original manuscript, the theory of domain wall conduction was developed for an extended DW. We made this point very clear in the revised manuscript. In the Supplementary Material we start by deriving “Tunneling in the model of zero length hDW”, then proceed with “Ballistic domain wall of finite length with scattering at the ends”, “Account for tunneling between the same spin modes in the polarized region”, and “General case of the domain wall of finite length” (the last section is new in the revised manuscript). In the “Summary of charge, spin and neutral currents in the absence of spin flip processes” we show that all these models result in the same measurable currents outside the domain wall.

As correctly noted by the Reviewer, hyperfine coupling is present in GaAs and we observe a buildup of nuclear polarization at high bias currents (see our answer below to the “experimental” part of the question). In the Supplementary materials we added a section “Effect

of spin flip processes”. There we discuss how the current flow is modified in the presence of spin-flip scattering that enables tunneling between quasiparticle modes with opposite spin polarization inside the domain wall. Current carried by a domain wall decreases with an increase of the spin-flip probability r , explicit dependence of currents on r is now derived in Ch. J of the Supplementary Materials. Experiments at large currents, where we observe build-up of the Overhauser field and where we expect that a finite nuclear polarization in the vicinity of the domain wall would lead to the saturation of spin-flip processes, show increase of the DW resistivity, in agreement with the developed theory.

The backscattering current decreases with increasing the helical channel length at a relatively long characteristic length of $L_0 = 47 \mu\text{m}$. Whereas this mechanism is not clearly stated in the text, this may involve a spin-flip process in the helical channels. If so, this characteristic length may change with nuclear spin polarization. The authors could have distinguished the effects on L_0 and i_{DW}^0 by repeating similar high-bias measurements for several lengths. Otherwise, the two possibilities should be described in the text. The authors applied a large current to polarize nuclear spins. However, this is not demonstrated because the large current may enhance backscattering in the bulk region. The authors should provide evidence of nuclear spin polarization, for example, long-term hysteresis associated with slow nuclear relaxation or NMR response.

The reviewer is absolutely correct that application of large dc current leads to nuclear polarization, as can be seen in the figure below. The non-equilibrium nuclear polarization is

FIG. R1. Resistance measured across the domain wall before (left) and after a dc current of 1 nA was applied for 40 min (right). Boundaries of spin transitions are shifted due to the nuclear Overhauser field (approximately +80 mT under gate G2).

long lived and slowly decays within ≈ 1 hour, consistent with previous experiments on 2D

gases. What we found, though, is that electron-nuclear angular momentum transfer is not a simple hyperfine interaction but can be mediated by spin waves. This is an interesting topic and will be a subject of a separate publication.

Position of spin transition boundary is a sensitive probe of nuclear polarization, and for this manuscript we present data close to thermal equilibrium, where small ac current does not affect position of the spin transition boundary. The enhanced resistance shown in the inset in Fig. 2 of the manuscript is measured right after application of a large dc current and before a build-up of nuclear polarization results in a measurable shift of spin transition boundaries. We modified the caption to Fig. 2 in the main text to include the following statement “In the inset i_{DW} dependence on large external dc current is plotted for $7 \mu\text{m}$ hDW for up and pu gates configuration right after the dc current is applied and before a measurable build-up of nuclear polarization.”

Reply to Reviewer #2:

The Reviewer finds that “The paper is well written and follows a clear experimental logic as well as a subsequent theoretical explanation.” In the following we hope to alleviate Reviewer’s concern that “the measured current is indeed coming from such a helical domain wall and not from a more trivial mechanism.”

For that I would ask for the measurement of the evolution of the longitudinal resistance $R_{12,34}$ when both gates (G1 and G2) are connected to the same voltage source. This would indicate how transparent is the constriction.

In the revised Supplementary Materials we included a new section “Device characterization”. In this section we include magnetoresistance scans at zero and non-zero gate voltages (Fig. S2), evolution of FQHE states and the spin transition in the vicinity of $\nu = 2/3$ as a function of gate voltages at different fields (S3), calibration of density vs gate voltage (S3), visualization of how we tune spin transitions to occur in the middle of the $\nu = 2/3$ state (S4), temperature dependence of resistance in the vicinity of the spin transition and excitation gaps in polarized and unpolarized states (S5), and resistance of the hDW for opposite field directions (S6).

Can the authors provide the same colorplot (R , ν , $\Delta \nu$) for opposite magnetic fields on the same sample? Contacts (especially for small ones) can have a huge impact on the measured data. Having opposite chirality can eliminate partly such doubts

This data is now included as Fig. S6 of the Supplementary Material.

Also is the tendency with length the same if the author takes the highest signal they get (in the green region) on the color plot (so not around the red circles). Density variations in the Hall bar can shift the real filling factor from the one calculated.

For the data in Fig. 2c we averaged data within ± 1 mV from the points marked on Fig. 1d with red circles. These points are not a "calculated", they are simply the experimentally determined centers of the u and p regions for the particular plot where 2D resistance is minimum and does not contribute substantially to the backscattering (yellow dashed lines on the plot). The gaps in the u and p regions are now plotted in Fig. S5 of the Supplement.

Did the authors measure the evolution of the measured current as a function of the magnetic field. I believe that by tuning the gate they probably can measure it at a much higher magnetic field which would make such states also more robust.

We include Fig. S4 where we show how boundaries of spin transitions evolve with magnetic field. There is a very narrow range of fields where reported experiments can be performed.

Does the author measure the Fig2c Inset at higher DC current than the one plotted on the graph? A clear saturation of the i_{DW} would give more credit to the theoretical model developed.

Yes, measurements were performed at higher currents as well and we found that resistance for $I_{dc} > 1$ nA is gradually decreasing. It should be noted that already at $I_{dc} = 1$ nA the Hall voltage $V_y = I_{dc}(3/2)h/e^2 = 39 \mu\text{V}$. At high Hall voltages u and p states are not insulating and some current flows through excited states in the bulk resulting in the reduction of the measured i_{DW} . Therefore we believe that inclusion of data for $I_{dc} > 1$ nA will be misleading.

Does the authors did any temperature dependence measurement of the current measured

This data is now included as Fig. S5 of the Supplementary Material.

Some additional remarks:

In Fig c the vertical arrow is not explained in the caption so we understand the meaning only at the end of the paper, Luttinger is sometimes named Luttenger, On Fig1 a) a clearer scale bar would help to understand the dimensions. It took me time time to understand that “The bar is 7 micron”; does not refer to the Hall bar but to the scale bar, On Fig1 d in the colorbar scale the integer 3 took the liberty to appear twice.

We corrected all this typos and omissions in the revised manuscript.

Reply to Reviewer #3:

The Reviewer “applauds the authors for undertaking such a study and find the results quite intriguing” although he/she concludes that at this point he/she “can understand neither the experimental findings nor the theory that supposedly explains those.” Judging from very legitimate questions raised by the Reviewer we take it as a personal challenge to improve the manuscript to make it more acceptable to a reader. Below we answer Reviewer’s questions and identify changes in the manuscript which hopefully clarify experimental and theoretical parts.

Let me begin with the experimental part. First and foremost, how do we know that the two halves of the sample are in the desired FQH states to begin with? I presume there is some evidence pointing to that but I cannot seem to find it in the paper. I am not even sure we are presented with the evidence that both sides of the device are at $\nu = 2/3$. E.g., Figure 1 presents some data on R_{45} and R_{34} , but all three leads appear to be attached to the same bottom edge (on both sides of the constriction). The relation of the measured resistances to the standard two- or four-contact measurements is unclear to me. Specifically, the vanishing R_{34} indicates that both halves of the device are sharing the same charge mode but beyond that, how do we know what state(s) this mode belongs to? Figure S2 of the Supplementary Materials *might* contain some evidence pointing in the right direction but even if so, it

is indirect and is not articulated in any easily accessible form. What I would ideally like to see is some “conventional” transport data indicating both the nature and the quality of the underlying FQH states. Specifically, would it be possible to open the constriction and gate the device uniformly first, demonstrating a bulk $\nu = 2/3$ state across the entire device through the conventional transport? Then, ideally, I would like to see a transition into a different $\nu = 2/3$ state controlled by the electrostatic gating. One should ideally be able to see transport signatures of a transition between the two states, as the bulk gap would vanish. While my request may be too ambitious (it is not obvious to me why it should be, but I am willing to give experimentalists the benefit of the doubt), *some* characterization along these lines should be presented.

In the revised Supplementary Materials we included a new section “Device characterization” which should answer all the question of the Reviewer #3 including R_{xx} and R_{xy} data for 2D gases under both gates where spin transition within the $\nu = 2/3$ state is clearly seen as a gap closure. We also want to note that in Fig 1 we show image of a ”typical” device, each sample has several such devices with different lithographically defined constrictions and contacts configurations which allow full characterization of a 2D gas under each gate. We want to stress that the length of the hDW is lithographically defined and cannot be controlled by the gates, gate voltages only control a gradient at the gate boundary which may or may not result in the formation of the hDW.

Beginning with establishing a uniform state across the entire device and characterizing it also crucial for understanding the role the constriction plays in this experiment. A study of tunnelling across the constriction in the absence of any domain walls would undoubtedly help separate transport signatures that are unique to the formation of a domain wall across the constriction. The issue of tunnelling across the constriction brings me to another gripe: how is the alleged length of the domain wall L determined experimentally? Since it is formed by electrostatic gating, it would be good to understand how the corresponding gate voltage is converted into microns that are used in Fig. 2(c). (On a related note, it would be helpful to see L marked on one of the figures, perhaps Fig. 1(a).)

The lack of characterization of the device also makes me wonder how the authors can rule

out an emergence of some compressible strip between the two quantum Hall states (instead of a microscopically sharp helical domain wall)?

These questions are now answered in Figs. S2-S6 of the Supplementary Material. We modified Fig. 1a to enlarge the gate boundary region where the hDW is formed. Lithographical size of the constriction defines the hDW length L (we ignore effects of a 2D gas depletion of $\sim 0.1 \mu\text{m}$ near mesa boundaries). There is no electron transfer across the constriction in the middle of the u and p phases when density is tuned to be uniform across the sample, the excitation gaps are $> 200 \text{ mK}$ as is shown in Fig. S5.

Turning to the theoretical description of the underlying physics, I must say that I also find it wanting. I do not find the presentation particularly helpful and in places it becomes outright confusing. To begin with, the authors jump between composite fermion (CF) and hierarchical pictures making it hard to follow. Specifically, the spin-polarized and unpolarized $\nu = 2/3$ states are described in terms of CF's in both the Introduction and Experimental Results section after which the paper jumps to the hierarchical description in the Theory section. Worse yet, the very first equation describing the action for the hierarchical edge of the polarized state is either wrong or at best mischaracterized in the surrounding texts. Setting aside the wrong overall sign, the action in Eq. (1) is clearly written for the K-matrix of the form $\text{diag}(1,-3)$, which describes two counterpropagating *charge* modes in the standard hierarchy construction, not “separated charged and neutral modes”, as stated in the paper. Compare this equation to Eqs. (1) as well as (7)-(8) of the seminal paper by Kane, Fisher & Polchinski, (Phys. Rev. Lett. 72, 4129 (1994)) outlining the phenomenon of spin-charge separation on such an edge. Curiously, the aforementioned paper is not even cited here, and nor is any other paper by Kane & Fisher, see e.g. their chapter “Edge state transport” in “Perspectives in Quantum Hall Effects” (1996), doi:10.1002/9783527617258.ch4. Just to be clear, I am not trying to make the authors add any citations here; I merely find the lack of these very relevant citations combined with the wrong/confusing statements rather worrisome. And it is not as if the authors felt pressured by the size constraints: in the same section they digress and discuss the inclusion of superconductivity, which is completely irrelevant to the subject at hand. My point is that the “Theory” section in its present form has hardly helped me understand the physics being studied here

(This unfortunate impression is further reinforced by relatively minor, yet adding-up things like a missing spatial derivative in the second term in Eq. (1), mismatching parentheses in Eq. (S2) or missing parentheses in Eq. (S6).)

We are grateful to the Referee for noticing a wrong sign in Eq. (1) and, correspondingly in Eq. (S18). These were typos, all our calculations were done with the correct (+) sign. We rectify these oversights and several other typos, including signs and a lost derivative in Eq. (1), in the revised manuscript. We have to admit that disappearance of the reference to Kane, Fisher and Polchinsky in our manuscript is an unfortunate accident and are happy to rectify this oversight. We add a few more references, including a reference to the review by Kane and Fisher, and a reference to Fendley, Ludwig and Saleur. Furthermore, we now fill in the presentation gaps that occurred in our first submission, which allows to explain appropriateness and novelty of our approach. We answer referee comments point by point:

1. Question on composite fermion, hierarchical, and quasiparticle pictures, pictures of separated charge and neutral modes, and action Eq.(1)

To clarify this point and improve the presentation, we include the discussion of the relation between composite fermion and separated charge and neutral modes in the main text. A conventional starting point for chiral edge states description are edge states originating from two filled Λ -levels of composite fermions with equal or opposite spin in the u and p phases, correspondingly, Eq. (S1). As shown in the Supplementary Materials, the description in terms of separated charged and neutral modes for the p phase Eq. (S18) is equivalent to the composite fermion description of the p phase edges Eq. (S1). Analogously, action Eq. (S1) leads to the transformed action Eq. (1) of the main text, which in the resubmitted version is written for the u phase. The actions Eq. (S18) and Eq. (1) both lead to separation of modes and coincide with the Kane, Fisher and Polchinsky action expressed in terms of the charge and neutral fields, under the condition that no electron scattering occurs, thereby precluding the composite fermion tunneling between the different modes. The exclusion of scattering is appropriate inside the u phase because scattering requires spin flips that are absent, and, as shown by Wu, Sreejith and Jain, Phys. Rev. B, **86** 115127 (2012), the u phase exhibits spin-charge separation with pure spin and pure charge modes moving in

opposite directions, as described by Eq. (1). To treat both phases, which exhibit quantization of Hall resistance $3/2 h/e^2$, on equal footing, we assume that no scattering between different modes occurs also in the p phase. At the same time, we show in Supplementary Materials that scattering between p phase CF modes inside the domain wall does not alter measurable currents. According to Kane, Fisher and Polchinsky, in four-terminal experiments the $3/2$ quantization of Hall resistance is only possible if the charge and neutral (spin) fields are separated, i.e, there is no coupling between them. This is precisely the feature of our Eqs. (1) and (S18). The reason for no coupling and the separation of modes in both phases in the absence of scattering on impurities, as discussed by Wen [Adv. Phys. 44, 405 (1995)], is the long-range Coulomb interaction, which makes equal the charge density couplings inside both CF modes along the edge, see Supplementary Materials, comment after Eq. (S17). As the description using separated charge and neutral (spin) modes and the composite fermion descriptions are equivalent, we use them interchangeably.

The sloppiness of the presentation is also evident in writing: “level crossing” requires an article in English; the QH discoverer’s name is *von* Klitzing, not Klitzing, and in any case, purely grammatically, it is either “the von Klizing constant” or “von Klitzing’s constant” (no “the”). Besides, it is simply beyond me why one would use R_q (and call it “the von Klizing constant”) instead of simply using h/e^2 , requiring no explanation or commentary of any kind.

We introduced corresponding corrections in the revised manuscript.

REVIEWERS' COMMENTS

Reviewer #1 (Remarks to the Author):

In my opinion, the paper is satisfactorily revised. I recommend the paper for publication in Nature Communications.

Reviewer #2 (Remarks to the Author):

The authors answered partially my questions about the characterization of the Hall bar by providing additional data. I am now a bit more convinced the measured signal does not come from a basic backscattering mechanism. I am a bit surprised that the authors had already a substantial contribution from the bulk if they overpass 1 nA. It has been reported to induce current as high as 40 nA and stay in a dissipationless state (PRL 116, 136804 (2016)).

This shows how fragile are the states in the studied samples (narrow Quantum well leading to higher magnetic field transition of these states would have been far more suitable). The measured data goes in the direction of transport through helical domain walls but I am personally not fully convinced seeing the experimental results. The exponential fit of i_{DW} over half a decade is clearly not convincing and the fact that i_{DW} goes down after 1 nA do not really reinforce the interpretation of the authors.

Nevertheless I would like to stress that such samples and experiments are very challenging to do and that it seems that the authors did the maximum to understand the physics at play.

Reviewer #3 (Remarks to the Author):

I am largely satisfied with the changes made by the authors in response to my and other referees' criticisms. I believe the paper should be accepted but I feel there are still some issues with the clarity of presentation, particularly that of the underlying theory.

Specifically,

- Theoretical presentation is still, in my opinion, wanting. Specifically, the authors use the term "hierarchical" loosely – to the point of being incorrect – and this is what tripped me the first time around. Unfortunately, the revisions did not fully address that issue. Specifically, the sentence introducing Eq. (1) states that the action "for $\nu = 2/3$ hierarchical edge states, e.g., for the u phase reads ..." This is misleading at best. The unpolarized state described by Eq. (1) is *not* a hierarchical state. A hierarchical state is a *polarized* state that would correspond to an identical action with the important difference: both fields in Eq. (1) would be charged! (This is something I had already pointed out in my original report). On the other hand, the unpolarized state can be thought of a "two-layer" state (with the "layer" corresponding to the spin) whose edge action can be transformed to the form given by Eq. (1) by the linear transformation presented in the preceding paragraph. Calling it "hierarchical" serves no other purpose but to confuse the reader. After all, the references by Kane et al. that I had brought to the authors' attention make a clear distinction between the action given by Eq. (1) for the *charged fields* and subsequent action for the renormalized fields.

A clearer language, consistent with the terminology used in the field, would go a long way in helping the reader avoid such unnecessary confusion.

- When discussing the tunnelling between polarized (p) and unpolarized (u) phases, the main text correctly states that this tunnelling is "carried by electrons" – see the sentence immediately preceding Eq. (2). In contrast, Section IV.D of the Supplementary Materials talks about "tunneling [SIC] between polarized and unpolarized phases for modes carried by composite fermions" while introducing the same (up to notations – more on this later) tunnelling Hamiltonian (S27). The electron and the composite fermion are different physical objects and the question about the physical nature of tunnelling is not as moot as it may seem in light of the identical (the notational discrepancies notwithstanding) expressions used to describe it. Specifically, only electrons are physically well-defined outside of an incompressible QH state. While one may choose to formally attach flux to electrons in a compressible state outside of the QH liquid, there is no sense in which such an object is a meaningful quasiparticle in such a state (if only because the charge and flux density need not locally match). This is not a "cosmetic" issue that one can just brush away: the issue of whether composite objects can actually tunnel had been addressed in a paper by Feng et al., "Suppression of the Josephson effect by quantum fluctuations in the fractional quantum Hall state", Phys. Rev. B 50, 11045 (1994), with the answer turning out to be negative (for the composite bosons in that case, but the underlying logic remains). The reason is basically that in the absence of a

physical mechanism suppressing quantum fluctuations outside of an incompressible QH state, the tunnelling of composite objects becomes essentially impossible.

Once again, this consideration does not seem to affect the description of tunnelling in this paper yet the accompanying statements should still be physically sound.

- Finally, the notations. It certainly appears that the authors could not settle on uniform notations and kept switching between them at will. Is it really too much to ask that a *resubmitted* manuscript is actually carefully read by the authors themselves before its submission?

Specifically, could the authors stick to the order of indices of the fields (i.e. $\alpha=\{p,u\}$ and $\beta=\{c,n/s\}$ according to the definition following Eq. (2) of the main text)? Eq. 2 itself clearly has the order of indices backwards (with the rest of the paper seeming using the declared notations). Also, the authors should also decide whether the numerical index (1 or 2) should precede (as in Eq. (2) and the main text) or follow (as in Sections IV.D and IV.E of the Supplementary Materials) and consistently stick to their choice throughout the manuscript.

Also, are \tilde{t} in Eq. (2) and \overline{t} in Eq. (S27) the same coupling constant (it certainly seems so)?

Such notational discrepancies are clearly not a sign of a thoughtful presentation and as a referee (and a potential reader) if find those instances off-putting, particularly given the fact that the manuscript has already undergone one round of revisions.

Given the potential importance of the experimental findings, I would recommend this manuscript for publication but I would request that the authors finally clean their theoretical presentation.

Reply to Reviewer #1:

In my opinion, the paper is satisfactorily revised. I recommend the paper for publication in Nature Communications.

Reply to Reviewer #2:

The authors answered partially my questions about the characterization of the Hall bar by providing additional data. I am now a bit more convinced the measured signal does not come from a basic backscattering mechanism. I am a bit surprised that the authors had already a substantial contribution from the bulk if they overpass 1 nA. It has been reported to induce current as high as 40 nA and stay in a dissipationless state (PRL 116, 136804 (2016)).

This shows how fragile are the states in the studied samples (narrow Quantum well leading to higher magnetic field transition of these states would have been far more suitable). The measured data goes in the direction of transport through helical domain walls but I am personally not fully convinced seeing the experimental results. The exponential fit of i_{DW} over half a decade is clearly not convincing and the fact that i_{DW} goes down after 1 nA do not really reinforce the interpretation of the authors.

Nevertheless I would like to stress that such samples and experiments are very challenging to do and that it seems that the authors did the maximum to understand the physics at play.

We glad that we were able to answer most of the Referee's questions. The question of the maximum DC current is an important one, however, we do not agree that quantitative comparison with the cited PRL 116, 136804 (2016) paper is justified. In the cited manuscript there is no local control of domain wall formation, and unknown number of domains and domain walls are formed. Thus, the maximum currents are expected to be *a priori* higher compared to a single conducting domain wall formed in our experiments. From this geometrical consideration, our results are *consistent* rather than in a disagreement with the results of the cited manuscript.

Reply to Reviewer #3:

I am largely satisfied with the changes made by the authors in response to my and other referees' criticisms. I believe the paper should be accepted but I feel there are still some issues with the clarity of presentation, particularly that of the underlying theory. - ...

Specifically, the authors use the term “hierarchical” loosely – to the point of being incorrect – and this is what tripped me the first time around. Unfortunately, the revisions did not fully address that issue....A clearer language, consistent with the terminology used in the field, would go a long way in helping the reader avoid such unnecessary confusion.

We thank the referee for clarifying his advice on not using terminology “hierarchical states” for cases that are not hierarchical despite being described by similar equations. We followed the advice and used a clear language and right terminology.

- When discussing the tunnelling between polarized (p) and unpolarized (u) phases, the main text correctly states that this tunnelling is “carried by electrons” – see the sentence immediately preceding Eq. (2). In contrast, Section IV.D of the Supplementary Information talks about “tunneling [SIC] between polarized and unpolarized phases for modes carried by composite fermions” while introducing the same... tunnelling Hamiltonian (S27). The electron and the composite fermion are different physical objects and the question about the physical nature of tunnelling is not as moot as it may seem in light of the identical (the notational discrepancies notwithstanding) expressions used to describe it. Specifically, only electrons are physically well-defined outside of an incompressible QH state. this consideration does not seem to affect the description of tunnelling in this paper yet the accompanying statements should still be physically sound.

In the revised version we now have the same description of electron tunneling in the Supplementary Information as we had used in the main text, both descriptions physically sound.

- Finally, the notations. It certainly appears that the authors could not settle on uniform notations and kept switching between them at will. Is it really too much to ask that a *resubmitted* manuscript is actually carefully read by the authors themselves before its submission?...

We carefully checked all notations and made them uniform throughout the main text and the Supplementary Information, and corrected typos. We changed notations on figures in accord with the changes in the text.

Given the potential importance of the experimental findings, I would recommend this manuscript for publication but I would request that the authors finally clean their theoretical presentation.

The theoretical presentation is now accurate and fully consistent.